# Exogenous Melatonin Reprograms the Rhizosphere Microbial Community to Modulate the Responses of Barley to Drought Stress

**DOI:** 10.3390/ijms23179665

**Published:** 2022-08-26

**Authors:** Fan Ye, Miao Jiang, Peng Zhang, Lei Liu, Shengqun Liu, Chunsheng Zhao, Xiangnan Li

**Affiliations:** 1Key Laboratory of Mollisols Agroecology, Northeast Institute of Geography and Agroecology, Chinese Academy of Sciences, Changchun 130102, China; 2College of Advanced Agricultural Sciences, University of Chinese Academy of Sciences, Beijing 100049, China; 3Key Laboratory of Agricultural Soil and Water Engineering in Arid and Semiarid Areas, Ministry of Education of China, Northwest A & F University, Yangling 712100, China; 4CAS Engineering Laboratory for Eco-Agriculture in Water Source of Liaoheyuan, Chinese Academy of Sciences, Changchun 130102, China

**Keywords:** rhizosphere microbiome, melatonin, drought, carbohydrate metabolism, *Hordeum vulgare*

## Abstract

The rhizospheric melatonin application-induced drought tolerance has been illuminated in various plant species, while the roles of the rhizosphere microbial community in this process are still unclear. Here, the diversity and functions of the rhizosphere microbial community and related physiological parameters were tested in barley under the rhizospheric melatonin application and drought. Exogenous melatonin improved plant performance under drought via increasing the activities of non-structural carbohydrate metabolism enzymes and activating the antioxidant enzyme systems in barley roots under drought. The 16S/ITS rRNA gene sequencing revealed that drought and melatonin altered the compositions of the microbiome. Exogenous melatonin increased the relative abundance of the bacterial community in carbohydrate and carboxylate degradation, while decreasing the relative abundance in the pathways of fatty acid and lipid degradation and inorganic nutrient metabolism under drought. These results suggest that the effects of melatonin on rhizosphere microbes and nutrient condition need to be considered in its application for crop drought-resistant cultivation.

## 1. Introduction

Soil microorganisms participate in rhizosphere nutrient cycling, which is closely related to plant growth and performance under abiotic stress [1,2,3]. The rhizosphere generally refers to the region of the soil controlled by roots of living plants, and enriching nutrients secreted from the roots. It is an important communication site for plant and microbes, showing important effects on crop production [4,5,6]. The rhizosphere microbes, especially some plant growth-promoting rhizobacteria (PGPR) and mycorrhizae fungi, are involved in the plant response to abiotic stress [7,8]. For instance, the *Bacillus aquimaris* can regulate osmotic equilibrium and ion homeostasis to stimulate plant growth by producing some extracellular secretions, such as hormones, polysaccharides and volatile organic compounds under salt and drought stress [9]. In addition, arbuscular mycorrhizal fungi (AMF) can promote plant water acquisition through the aids of the extraradical hyphae system to cope with drought [10,11].

As one of the abiotic stresses, drought negatively impacts the water relations in plants, exemplifying the reduced relative water content and the transpiration in leaves [12,13,14]. More importantly, reduced transpiration under drought stress alters the leaf microenvironment, causing damage to cell membranes, leading to intracellular ion disorders and accumulation of reactive oxygen species (ROS) [15,16] This then results in oxidative damage to chloroplasts and a reduction in chlorophyll content, which limits photosynthesis [13,17]. As affected by drought, soil microbiota suffer disturbance due to the changes in soil moisture, physical, and chemical properties [18,19]. Drought has a significant impact on the structure and function of the soil microbial community, normally reducing the abundance and diversity [20,21]. The rhizosphere microbial community is recruited from the soil and strongly influenced by host plants, and the recruitment of rhizosphere microbiota is easily affected by drought. In sorghum and rice, it has been documented that drought delays the development of the root microbiome, while long-term drought causes irreversible impacts on root microbiome development [22].

As an indoleamine, melatonin (*N*-acetyl-5-methoxytryptamine) is a pleiotropic signal molecule with various physiological functions in plants, microorganisms, and animals [19]. Melatonin benefits plant performance under abiotic and biotic stress [23,24]. Exogenous melatonin alleviates photosynthetic inhibition and oxidative damage [25,26,27], and regulates carbon assimilation, hence improving drought tolerance [28,29,30]. This process is involved in the signaling pathways of abscisic acid (ABA) and salicylic acid [31,32]. Previous studies illuminated that melatonin has direct effects on microbiota in animal guts and soil [33,34]. In plants, the rhizospheric melatonin application may regulate the structure and function of the microbial community in the plant rhizosphere, hence influencing the plant’s responses to drought.

To explore the roles of melatonin induced reprograming of the rhizosphere microbial community in barley drought response, the metabolisms of ROS, carbohydrate, and melatonin in plants and the compositions of bacterial and fungal communities in plant rhizosphere were tested. The objectives were (1) to evaluate the effects of the rhizospheric melatonin application on plant performance under drought and (2) to investigate the effects of melatonin on bacterial and fungal communities and their association with the barley plant performance in response to drought.

## 2. Results

### 2.1. Physiological Characteristics, ROS and Carbohydrate Metabolism Enzyme Activities as Affected by Melatonin and Drought

Exogenous melatonin alleviated the wilting of barley leaves under soil water deficit (Appendix A). Drought significantly increased the H_2_O_2_ concentration in barley leaves compared with the non-stress control, and the application of exogenous melatonin significantly alleviated the accumulation of H_2_O_2_ in leaves under drought stress (Figure 1A). The concentrations of proline and glycine betaine in barley leaves were also obviously increased by drought. However, the melatonin application increased the glycine betaine concentration, while decreasing the proline concentration in leaves under drought stress (Figure 1A,B).

Compared with the non-stress control, drought significantly increased the activities of AGP, UGPase, and FK, while significantly decreasing the two invertases (cytInv and vacInv) in roots (Figure 2A and Appendix A). Under drought, rhizospheric melatonin significantly increased the activities of AGP, UGPase, PGI, Ald, HXK, PFK, cytInv and vacInv, while decreasing the FK activity in roots. In barley leaves, drought increased the activities of FK and HXK, but decreased the PGI activity, in relation to the control. Melatonin significantly increased the activities of G6PDH and PGM, while remarkably decreasing the activities of FK, HXK, and PFK in leaves under drought. For the antioxidant enzyme systems, drought increased the activities of APX and GR, while decreasing the CAT activity in roots (Figure 2B and Appendix A). Under drought, melatonin pretreatment significantly enhanced the activities of CAT, APX, DHAR, GST, and MDHAR in roots. In barley leaves, drought resulted in decreased activities of SOD, APX, and GST, compared with the control. Under drought stress, melatonin treatment decreased the activities of APX and SOD, which was closer to the levels under the non-stress control.

### 2.2. Endogenous Melatonin Metabolism and ABA as Affected by Melatonin and Drought

Drought significantly decreased the activities of T5H and SNAT in barley roots, while significantly increasing the concentrations of serotonin and AFMK (Figure 2C and Appendix A). Under drought, the rhizospheric melatonin application obviously increased the activity of SNAT and the concentrations of *N*-Acetylserotonin and 2-Hydroxymelatonin, while decreasing the AFMK concentration in roots. Melatonin pretreatment significantly increased the endogenous melatonin level in roots under drought. In leaves, drought increased the TDC and ASMT activities, while decreasing T5H activity more than the control, resulting in a higher AFMK concentration while lower tryptamine and 2-hydroxymelatonin levels. Under drought, melatonin pretreatment decreased the activities of TDC and ASMT, resulting in deceased AFMK concentration, while increasing 2-Hydroxymelatonin concentration. In addition, drought significantly enhanced the ABA levels in leaves, roots, and soil; however, rhizospheric melatonin remarkably reduced the ABA concentrations under drought (Figure 3).

### 2.3. Rhizosphere Microbial Community as Affected by Melatonin and Drought

No significant difference in the Chao1 index and Shannon diversity of barley rhizosphere microbial communities was found between the normal control and drought treatments (Figure 4A,B). Under drought, the rhizospheric melatonin application decreased the Shannon diversity of bacterial community, while having no effect on that of the fungal community. In addition, melatonin did not influence the Chao1 index of either bacterial or fungal communities under drought. The beta diversity of the different microbial communities was presented using a principal coordinate analysis (PCoA) based on Bray–Curtis dissimilarity metrics (Figure 5). The microbial communities of these three treatments were clustered separately. The significance analysis of community structure differences (ADONIS test) indicated that drought and the rhizospheric melatonin treatment significantly changed the beta diversity of bacterial and fungal communities.

The dominant rhizosphere bacteria mainly included five phyla, i.e., Actinobacteria, Proteobacteria, Acidobacteria, Chloroflexi, and Gemmatimonadetes (Figure 4C). Compared with the non-stress control, drought increased the relative abundance of Actinobacteria phylum in barley rhizosphere regardless of melatonin treatments. However, the relative abundance of Proteobacteria phylum was decreased under drought stress in relation to the control, while the rhizospheric melatonin application had an opposite effect. Furthermore, drought reduced the relative abundances of Acidobacteria and Gemmatimonadetes in relation to the control. Under drought, the rhizospheric melatonin application resulted in a decline in Acidobacteria and Gemmatimonadetes abundances. No obvious change was found in the relative abundance of Chloroflexi phylum among these three treatments.

The dominant rhizosphere fungi mainly belonged to Ascomycota, Mortierellomycota, Basidiomycota, and Olpidiomycota (Figure 4D). The relative abundance of Ascomycota phylum accounted for the largest proportion. Drought significantly enriched Ascomycota in the barley rhizosphere and the rhizospheric melatonin application further enhanced the enrichment. Compared with the non-stress control, the relative abundances of Mortierellomycota and Basidiomycota was decreased by both melatonin and drought. Additionally, drought increased the relative abundance of Olpidiomycota compared with the control, while melatonin treatment caused a slight decrease in its abundance under drought.

### 2.4. Microbial Diversity as Affected by Melatonin and Drought

To identify the key taxon of rhizosphere microbiota among treatments, the linear discriminant analysis effect size (LEfSe) was used to detect the biomarkers with significantly changed relative abundances among treatments with an LDA threshold of 3.5 (*p* < 0.05) (Figure 6). These three treatments could be distinguished at the phylum level. Biomarkers most strongly associated with drought were from the Actinobacteria phylum; however, Proteobacteria and Patescibacteria phyla were enriched in the D + MT treatment (Figure 6A). Moreover, the biomarkers associated with drought were from the Thermoleophilia class, Solirubrobacterales order, and Gaiellales order, while that for the rhizospheric melatonin treatment were from Actinobacteria class, Gammatimonadetes class, Micrococcales order, and Propionibacteriales order.

The fungal biomarkers as affected by drought showed significantly different relative abundances at Glomeromycota phylum, Glomeromycetes class, Glomerales order, and Glomeraceae family (Figure 6B). After melatonin pretreatment, the significantly different fungal taxa were from Ascomycota phylum, Sordariomycetes class, Eurotomycetes class, Hypocreales order, Chaetothyriales order, Nectriaceae family, and Herpotrichiellaceae family under drought.

### 2.5. Prediction of Functional Composition in Microbial Community

The different functional capacities of microbial communities inferred by PICRUSt2 among treatments are shown in Figure 7. The rhizosphere bacterial functional types were from seven differentially abundant functional MetaCyc pathways at level 1, mainly being enriched in biosynthesis, degradation processes, and the generation of precursor metabolic (Figure 7A). The fungal functional types were enriched in biosynthesis, degradation processes, detoxification, glycan pathways, and metabolic clusters at level 1 (Figure 7B).

Melatonin treatment and drought significantly affected the functional MetaCyc pathways of the rhizosphere microbiome at level 2 (Figure 7). Compared with the control, drought significantly reduced the relative abundances of bacteria in the pathways of amino acid biosynthesis, nucleoside and nucleotide biosynthesis, respiration, and electron transfer, but increased that in fermentation, TCA cycle, carbohydrate degradation, and carboxylate degradation (Figure 7C). A similar trend was found in the combination of melatonin and drought. Furthermore, the fungal relative abundances of functional pathways were affected by drought to a lesser extent than that of bacteria in the barley rhizosphere (Figure 7D). The relative abundances of fungi related to the glyoxylate cycle, inorganic nutrient metabolism, and glycan biosynthesis was decreased by drought, in relation to the control. Melatonin treatment further reduced the fungal relative abundances in these pathways, except for the glycan biosynthesis. Under drought, the fungal relative abundance of nucleoside and nucleotide degradation increased in relation to the control, and its abundances further increased after the melatonin application.

## 3. Discussion

Non-structural carbohydrates provide sustained energy for plant growth, and are also closely associated with abiotic stress tolerance [35,36]. In the present study, the activities of enzymes catalyzing sucrose degradation (cytInv and vacInv) were inhibited, while that of the enzymes related to starch synthesis (AGP and UGPase) were enhanced by drought in barley roots. This resulted in the sucrose and starch accumulations in roots, which was consistent with the study in soybean [37]. Melatonin has been reported to regulate the carbohydrate metabolism to cope with abiotic stress in plants [38]. Here, the rhizospheric melatonin application improved the activities of glycolytic enzymes (PGI, Ald, HXK and PFK) in roots under drought, suggesting that melatonin helps to maintain the carbohydrate metabolism for a normal growth of barley under drought stress.

Melatonin pretreatment was effective for eliminating excessive H_2_O_2_ in barley leaves and reducing oxidative damage under drought stress [15]. The antioxidants, such as proline and glycine betaine, not only act as osmo-protectants but also contribute to alleviating osmotic stress [14,39]. Melatonin pretreatment increased the concentration of glycine betaine in barley leaves under drought stress. However, melatonin pretreatment decreased the proline concentration as compared to drought treatment alone. Huang and Imran et al. also found that the proline level was decreased in melatonin-pretreated maize and soybean under drought stress [17,40], which is in agreement with our result.

Abscisic acid acts as the core regulating pathway in plant drought responses, by promoting leaf stomatal closure and activating the antioxidant system to alleviate drought stress [41]. Here, drought increased the ABA concentrations, while rhizosphere treatment with melatonin decreased the ABA levels in both roots and leaves. In addition, a significant increase in endogenous melatonin concentration was found under the rhizospheric melatonin application. The antagonistic relationship between ABA and melatonin has been documented in the induction of stress tolerance in various plant species [42,43], which was related to that melatonin can regulate the synthesis and catabolism of ABA in plants [44,45,46]. It has been illuminated that the melatonin pretreatment can induce the accumulation of endogenous melatonin, hence improving plant cold tolerance [47]. In the present study, the rhizospheric melatonin application might also regulate the drought response by affecting the endogenous melatonin and ABA metabolisms in barley. In the process of rhizospheric melatonin-induced modulations of plant drought response, the antioxidant enzyme system could be a key regulator [48]. In this study, the activities of most antioxidant enzymes in roots were enhanced by melatonin under drought. Enhanced antioxidant enzyme activities have been reported to be an important approach for exogenous melatonin-induced stress tolerance [49]. Melatonin can eliminate excess ROS accumulation by promoting the ascorbate–glutathione (AsA-GSH) cycle [50,51], which is consistent with this study where the activities of DHAR, GST, and MDHAR were significantly enhanced by melatonin in roots.

Accumulated evidence shows that rhizosphere microbes can regulate plant performance under stress conditions [7,8]. For instance, short-term drought can reduce the abundance and diversity of the rhizosphere microbiome, while prolonged severe drought has irreversible effects on the rhizosphere microbiome [20,21]. Here, the α-diversity of barley rhizosphere bacterial and fungal communities were not affected by a 14-day progressive drought. It has been reported that the decrease in the soil microbial community diversity and abundance is indirectly caused by changes in soil pH and organic carbon content under drought [20]. Thus, a short-term drought may not be sufficient to affect the soil pH and organic carbon content, hence changing the abundance and diversity of the barley rhizosphere microbial community. It should be noted that drought significantly altered the β-diversity of barley rhizosphere bacterial and fungal communities, affected the composition of microbial communities, and caused changes in the metabolic functions of bacteria. Different microbial community compositions can reflect specific ecological functions and adapt themselves to environmental changes [52]. Drought has been reported to enrich several types of soil microbes, causing changes in functions of community [20,53]. Here, a significant enrichment of Actinomycetes in the barley rhizosphere was found under drought. It has been documented that drought tends to enrich Gram-positive bacteria, especially Actinomycetes, due to the high drought tolerance caused by the structure of Gram-positive bacteria themselves [16,18,54]. Furthermore, Actinomycetes play a major role in the cycling of soil organic matter and can inhibit the growth of several plant pathogens in roots [55]; thus, its enrichment in the rhizosphere was beneficial for barley plant growth. For the fungal community in the rhizosphere soil, drought increased the abundance of Ascomycota, which is consistent with the results in arid regions [56]. Ascomycetes in the rhizosphere are involved in the plant nitrogen uptake, which may be beneficial for plant nutrient status under drought [18,56,57]. In addition, Glomeromycota was found to be drought sensitive in the rhizosphere soil, which can form tufts of mycorrhizae with terrestrial plants [58,59], and this symbiotic structure assists plants in phosphorus uptake.

Melatonin is involved in microbiota reprogramming and microbial pathogen tolerance induction in animals and plants [60,61,62]. Here, exogenous melatonin affected the microbial community composition under drought. Compared with the non-stress control, melatonin significantly reduced the evenness of bacterial communities under drought, while having no effect on fungal communities. It has been found that the exogenous application of melatonin regulates the soil microbiome under salt stress [63]. Also, exogenous melatonin affects plant root growth and metabolite levels under drought [28,64,65], hence altering plant–microbe interactions and potentially reprogramming the rhizosphere microbiome. For example, in this study, Patescibacteria and Proteobacteria were sensitive to melatonin application. Patescibacteria, as autotrophic bacteria, is involved in nutrient transformation at the rhizosphere level [66]. In addition, the Proteobacteria phylum and betaproteobacteria order in the D + MT treatment were significantly different from other treatments, which included some species with functions of nitrogen fixation and ammonia oxidation [67]. In betaproteobacteria, the Burkholderiaceae were also regulated by melatonin under drought, which has been found to promote plant growth through nitrogen fixation [68]. In the biomarkers belonging to the Actinomycetes, as oligotrophic microbes [69], Micrococcaceae, Intrasporangiaceae, and Nocardioidaceae were also affected significantly by melatonin treatment, suggesting that melatonin could affect the trophic type of rhizosphere bacterial communities under drought. In addition, melatonin treatment caused an enrichment of Ascomycota under drought, and regulated the acidophilic fungal members Sordariomycetes and Eurotiomycetes class from Ascomycota. Based on the functions of these bacteria, it was suggested that melatonin may change nutrient status and promote the denitrification process in barley rhizosphere under drought.

Both drought and melatonin application significantly affected the functional abundance based on the prediction of PICRUSt2. The barley rhizosphere microbiome for different treatments focused on biosynthesis, degradation processes, and generation of precursors by the functional annotation of MetaCyc. The relative abundance of degradation metabolic pathways, including carbohydrate and carboxylate degradation, and the TCA cycle increased in the rhizosphere bacterial community by melatonin application under drought, indicating that melatonin stimulated the metabolism of the rhizosphere bacterial community to provide energy for plant growth. Compared with the non-stress control, melatonin and drought decreased the relative abundance of amino acids, nucleotides, and nucleotide biosynthesis in the bacterial community. Meanwhile, these treatments increased the degradation of nucleosides and nucleotides, while decreasing the degradation of fatty acids and lipids and the relative abundance of inorganic nutrient metabolic pathways and TCA cycle in the fungal community. These results suggest that melatonin reprograms the metabolic function of the barley rhizosphere microbiome, and the bacterial and fungal communities in the barley rhizosphere respond differently to melatonin treatment under drought.

## 4. Materials and Methods

### 4.1. Plant Material and Growth Condition

The seeds of spring barley cv. Steptoe were sterilized with 80% ethanol solution and 1% sodium hypochlorite solution and finally washed with sterile water. Four seeds were sown in each pot with 1 kg of clay soil. The soil was collected at the Northeast Institute of Geography and Agroecology, Chinese Academy of Sciences, Changchun, and passed through a sieve of 2 mm aperture. The organic matter, available nitrogen, available phosphorus, and available potassium were 22.60 g·kg^−1^, 141.3 mg·kg^−1^, 62.8 mg·kg^−1^, and 147.1 mg·kg^−1^, respectively. The plants were grown in a growth chamber at 26 °C/20 °C (day/night, 22,000 Lux/0 Lux, 12 h/12 h) with a relative humidity of 60 ± 5%.

### 4.2. Treatments

At the 6-leaf stage, the rhizospheric treatment with 2 mmol/L melatonin was applied every three days for 10 times on half of the barley plants (50 mL per plant), while the rest were irrigated with the same amount of water as the control (N). Three days after the last melatonin application, the melatonin-treated plants and half of the control plants were subjected to a progressive soil drying as the drought treatment, by withholding irrigation from the pots until the soil water content decreased to 30% of the pot water holding capacity. The normal control group was well watered to reach 85% of the pot water holding capacity. Six pots were included in each group. Therefore, three treatments were established: N, the normal control; D, drought treatment; D + MT, drought + 2 mmol/L melatonin pretreatment. The last fully expanded leaves and roots were collected for analysis after the drought treatment. The rhizosphere soil samples were collected using a 2.0 mm sterile sieve and snap frozen in liquid nitrogen and stored at −80 °C.

### 4.3. Concentrations of H_2_O_2_, Proline and Glycine Betaine in Barley Leaves

The H_2_O_2_ concentration was measured according to the method of Zheng et al. [70]. The leaf samples were homogenized in cold acetone, and then the extracts were determined using a hydrogen peroxide assay kit (Beijing Solarbio Science & Technology Co., Ltd., Beijing, China) and the absorbance was measured at 415 nm to quantify the H_2_O_2_ concentration. The proline concentration was measured using a proline assay kit (Beijing Boxbio Science & Technology Co., Ltd., Beijing, China). Proline in barley leaf was extracted by sulfosalicylic acid and reacted with acidic ninhydrin, and then its absorbance was measured at 520 nm to quantify the concentration of proline. The glycine betaine in barley leaves were extracted in 80% methanol and measured by a microanalysis method using a betaine assay kit (Beijing Boxbio Science & Technology Co., Ltd., China). The absorbance was measured at 525 nm to quantify the concentration of glycine betaine. 

### 4.4. Activities of Enzymes in ROS and Carbohydrate Metabolisms

The measurements of antioxidant enzyme activities were applied according to the methods of Fimognari et al. (2020) [71], including superoxide dismutase (SOD), catalase (CAT), cytoplasmic/apoplastic peroxidases (POX/cwPOX), ascorbate peroxidase (APX), glutathione reductase (GR), glutathione S-transferase (GST), dehydroascorbate reductase (DHAR), and monodehydroascorbate reductase (MDHAR). The determination of carbohydrate enzyme activities was conducted following the methods of Jammer et al. (2015) [72]. The activities of sucrose synthase (Susy), hexokinase (HXK), aldolase (Ald), phosphofructokinase (PFK), phosphoglucoisomerase (PGI), ADP- Glucose pyrophosphorylase (AGPase), phosphoglucomutase (PGM), fructokinase (FK), UDP-glucose pyrophorylase (UGPase), and glucose-6-phosphate dehydrogenase were determined in kinetic enzyme assays. The activities of invertases (cytoplasmic invertase (cytInv), vacuolar invertase (vacInv), and cell wall invertase (cwInv)) were tested in the endpoint assays. The measurement was performed with an Epoch Take3 spectrophotometer (BioTek Instruments, Inc., Winosky, VT, USA) with a 96-well microtiter format. Four replicates were included in each treatment.

### 4.5. Melatonin Synthesis and Metabolism Enzyme Activities and Related Metabolites

The analysis of activities of enzymes in melatonin synthesis and metabolism including tryptophan decarboxylase (TDC), tryptamine 5-hydroxylase (T5H), caffeic acid *O*-methyltransferase (COMT), *N*-acetylserotonin *O*-methyltransferase (ASMT), and serotonin *N*-acetyltransferase (SNAT), was conducted following the protocol of the enzyme-linked immunosorbent assay (ELISA) Kit (Shanghai Youxuan Biotech Co., Ltd., Shanghai, China). The melatonin metabolism related metabolites in roots and leaves were measured with an Epoch™ microplate spectrophotometer (BioTek Instruments, Inc., Winosky, VT, USA) by enzyme-linked immunosorbent assay (ELISA) kits, including melatonin, tryptophan, tryptamine, serotonin, *N*-acetylserotonin (NAS), 2-hydroxymelatonin (2-HM) and *N*^1^-acetyl-*N*^2^-formyl-5-methoxyknuramine (AFMK). Four replicates were included in each treatment.

### 4.6. Determine of ABA Concentration

The ABA concentrations in barley (root and leaf) and soil were measured with an Epoch™ microplate spectrophotometer (BioTek Instruments, Inc., Winosky, VT, USA) by using enzyme-linked immunosorbent assay (ELISA) kits (Shanghai Youxuan Biotech Co., Ltd., Shanghai, China). Four replicates were included in each treatment.

### 4.7. DNA Extraction and 16S/ITS rRNA Gene Sequencing

Microbial genomic DNA of rhizosphere soil (0.5 g) was extracted using the OMEGA Soil DNA Kit (D5625-01) (Omega Bio-Tek, Norcross, GA, USA). A NanoDrop ND-1000 spectrophotometer (Thermo Fisher Scientifific, Waltham, MA, USA) and agarose gel electrophoresis were used to test the quantity and quality of the extracted DNA, respectively. Each treatment consisted of four biological replicates.

PCR amplification of the bacterial 16S V3-V4 region was performed using forward primer 338F (5′-ACTCCTACGGGAGGCAGCA-3′) and reverse primer 806R (5′-GGACTACHVGGGTWTCTAAT-3′). The forward primer ITS5F (5′-GTGCCAGCMGCCGCGGTAA-3′) and the reverse primer ITS2R (5′-CCGTCAATTCCTTTGAGTTT-3′) were used simultaneously for the fungal ITS V1 region. The components of PCR amplification contained 5 μL of reaction buffer (5×), 5 μL of GC buffer (5×), 0.25 μL of Fast pfu DNA Polymerase (5 U/μL), 2 μL (2.5 mmol/L) of dNTPs, 1 μL (10 μmol/Lu) of each Forward and Reverse primers, 1 μL of DNA Template, and 9.75 μL of ddH_2_O.

The PCR reactions for microbial genomic DNA were performed using the following procedure: denaturation at 98 °C for 5 min, annealing at 52 °C for 30 s, extension at 72 °C for 1 min, and final extension at 72 °C for 5 min. The PCR products were purified using Vazyme VAHTSTM DNA cleaning beads (Vazyme, Nanjing, China) and quantified with the Quant-iT PicoGreen dsDNA detection kit (Invitrogen, Carlsbad, CA, USA). Paired-end sequencing (2 × 250) was performed using Illumina NovaSeq 6000 (Hayward CA the USA), and the purified amplicons were pooled equimolarly at Shanghai Personal Biotechnology Co., Ltd. (Shanghai, China). The sequences were quality filtered, denoised, merged and chimera removed using the DADA2 plugin [73]. The sequence standard operation procedure and data analysis were performed using the QIIME2 and R packages (v3.2.0). Unique reads with 100% similarity were clustered into ASVs (amplified subsequence variants) based on representative 16S or ITS1 sequences. Taxonomy was assigned to ASVs using the classify-sklearn naïve Bayes taxonomy classifier in feature-classifier plugin [74] against the SILVA Release132 (http://www.arb-silva.de) for bacteria, while UNITE Release 8.0 (https://unite.ut.ee/) accessed on 22 November 2020 for fungi Database [75]. Sequences from the host were filtered out from the bacterial ASV table.

### 4.8. Statistical Analysis

The physiological data was first tested for homogeneity of variance by boxplot, followed by a one-way ANOVA, and all data were tested for significant differences at the *p* < 0.05 level using SPSS 22.0 (SPSS Inc., Chicago, IL, USA).

## 5. Conclusions

Rhizospheric melatonin application improved barley drought tolerance by regulating nonstructural carbohydrate metabolism and activating antioxidant enzyme activity. Additionally, melatonin reshaped the rhizosphere microbiome. Under drought, the homogeneity of bacterial communities and the abundance of dominant fungal phyla were regulated by melatonin application, which were related to carbohydrate and carboxylate degradation, fatty acid and lipid degradation, and inorganic nutrient metabolism. Thus, it is suggested that the roles of rhizosphere microbes are of importance in the application of melatonin for crop production.

## Figures and Tables

**Figure 1 ijms-23-09665-f001:**
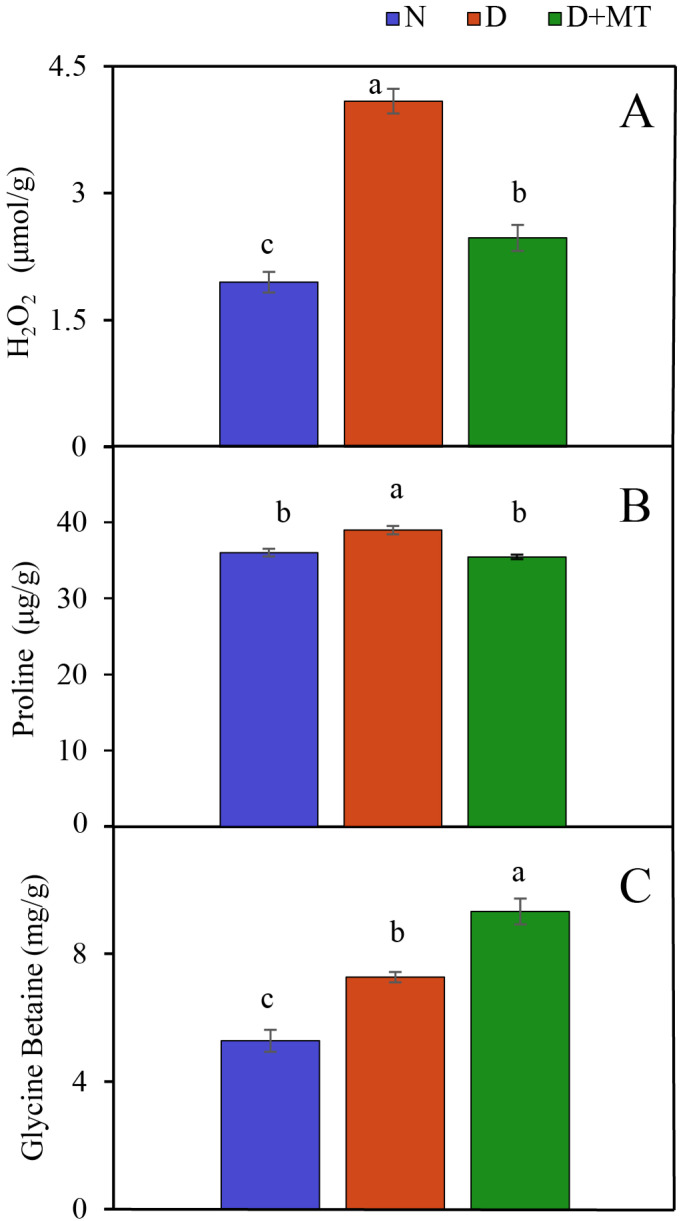
Concentrations of H_2_O_2_, proline and glycine betaine in barley leaves as affected by melatonin treatment and drought stress. The letters (**A**–**C**) corresponds to H_2_O_2_, proline and glycine betaine concentrations, respectively. Data are reported as mean ± SE (H_2_O_2_ and proline, *n* = 4; glycine betaine, *n* = 3). Different small letters indicate significant differences at the *p* < 0.05 level. N, normal control; D, drought stress; D + MT, melatonin + drought stress.

**Figure 2 ijms-23-09665-f002:**
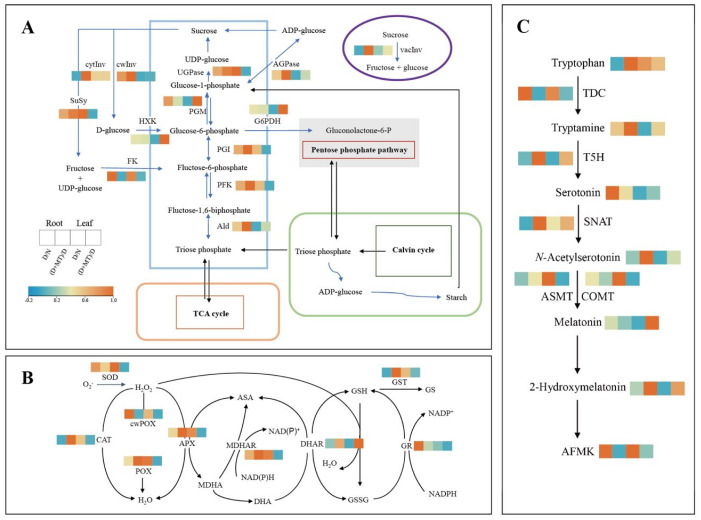
Metabolite concentrations and activities of enzymes involved in metabolisms of reactive oxygen species, carbohydrate, and melatonin in root and leaf as affected by melatonin treatment and drought stress in barley. (**A**) Change in carbohydrate metabolism enzyme activities. (**B**) Change in antioxidant enzyme activities. (**C**) Change in melatonin metabolism enzyme activities and concentrations of related metabolites. The difference of concentration/activity for a given metabolite/enzyme between the normal control (N) and treatments is normalized and converted to a color scale. An increase and decrease in concentration/activity are indicated in red and blue, respectively. N, normal control; D, drought stress; D + MT, melatonin + drought stress. SOD, superoxide dismutase; cwPOX, cell wall peroxidase; CAT, catalase; POX, peroxidase; APX, ascorbate peroxidase; MDHAR, monodehydroascorbate reductase; DHAR, monodehydroascorbate; GR, glutathione reductase; GST, glutathione S-transferase; cytInv, cytoplasmic invertase; vacInv, vacuolar invertase; cwInv, cell wall invertase; UGPase, UDP-glucose pyrophosphorylase; Susy, sucrose synthase; HXK, hexokinase; G6PDH, glucose-6-phosphate dehydrogenase; PGI, phosphoglucoisomerase; FK, fructokinase; PFK, phosphofructokinase; PGM, phosphoglucomutase; AGPase, ADP-glucose pyrophosphorylase; Ald, aldolase; T5H, Tryptamine 5-hydroxylase; AFMK, *N*^1^-acetyl-*N*^2^-formyl-5-methoxyknuramine; TDC, Tryptophan decarboxylase; SNAT, Serotonin *N*-acetyltransferase; ASMT, *N*-acetylserotonin methyltransferase; COMT, caffeic acid, *O*-methyltransferase.

**Figure 3 ijms-23-09665-f003:**
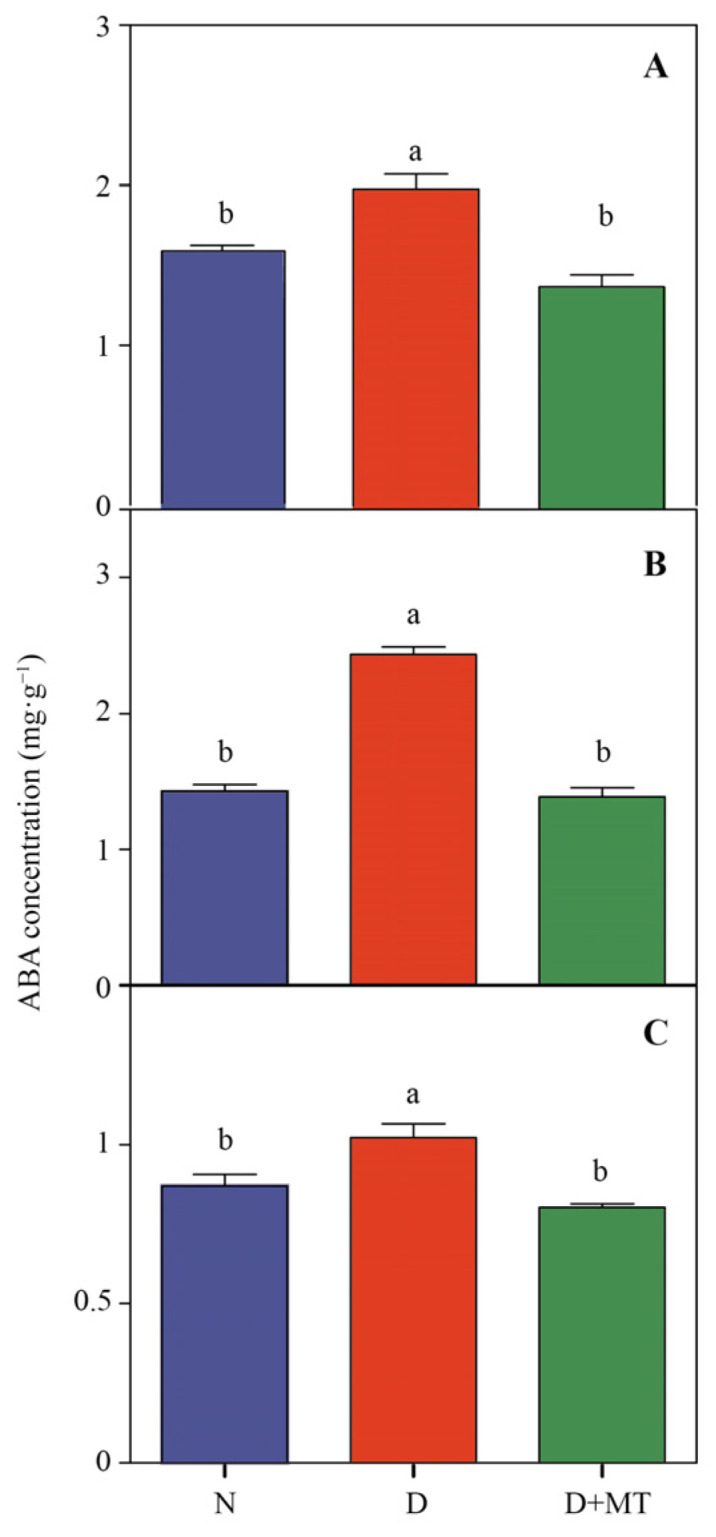
Concentrations of ABA in soil, barley root, and leaf as affected by melatonin treatment and drought stress. The letters (**A**–**C**) corresponds to leaves, roots, and soil, respectively. Data are reported as mean ± SE (*n* = 4). Different small letters indicate significant differences at the *p* < 0.05 level. N, normal control; D, drought stress; D + MT, melatonin + drought stress.

**Figure 4 ijms-23-09665-f004:**
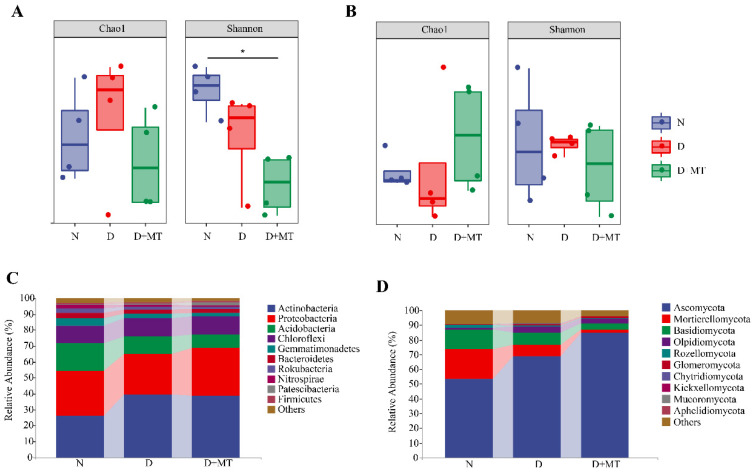
The α diversity and relative abundance in phylum level of bacteria (**A**,**C**) and fungi (**B**,**D**) as affected by melatonin treatment and drought stress. “*” indicates significance at *p* < 0.05. N, normal control; D, drought stress; D + MT, melatonin + drought stress.

**Figure 5 ijms-23-09665-f005:**
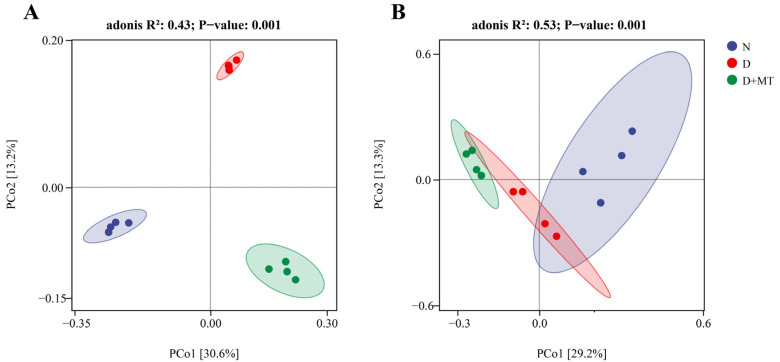
PCoA (Principal Co-ordinates Analysis) of Bray–Curtis distances (*p* < 0.05) for bacterial (**A**) and fungal (**B**) communities under the combination of melatonin treatment and drought stress. N, normal control; D, drought stress; D + MT, melatonin + drought stress.

**Figure 6 ijms-23-09665-f006:**
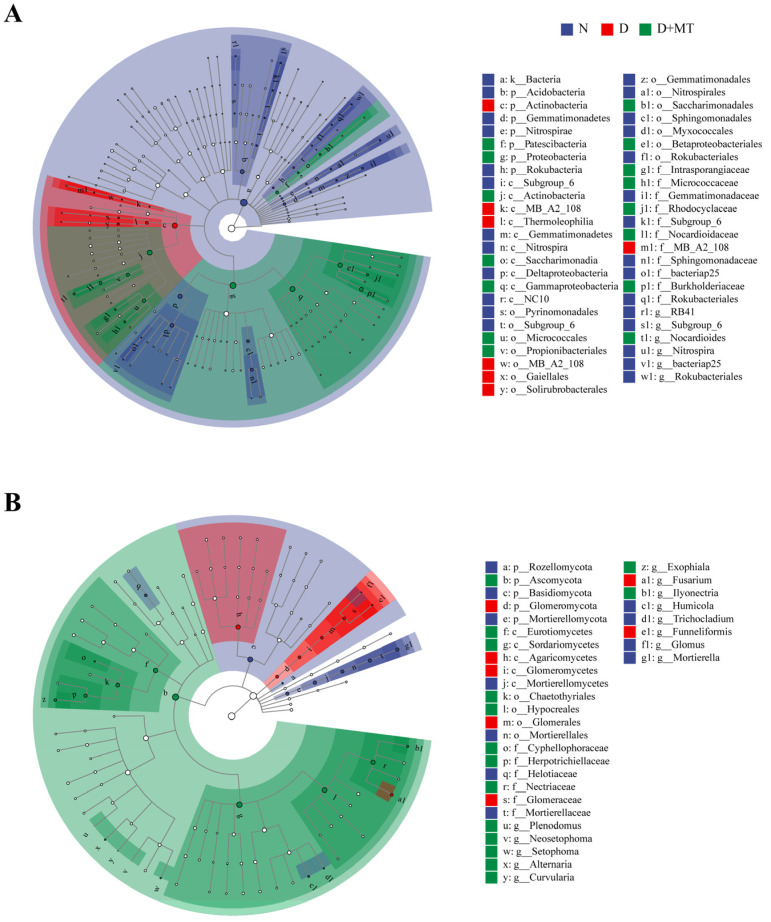
Linear discriminant analysis effect size (LEfSe) of the bacterial (**A**) and fungal (**B**) taxa with an LDA score > 3.5 under the combination of melatonin treatment and drought stress. Cladograms indicate the phylogenetic distribution of microbial lineages. Circles represent phylogenetic levels from kingdom to genus. N, normal control; D, drought stress; D + MT, melatonin + drought stress.

**Figure 7 ijms-23-09665-f007:**
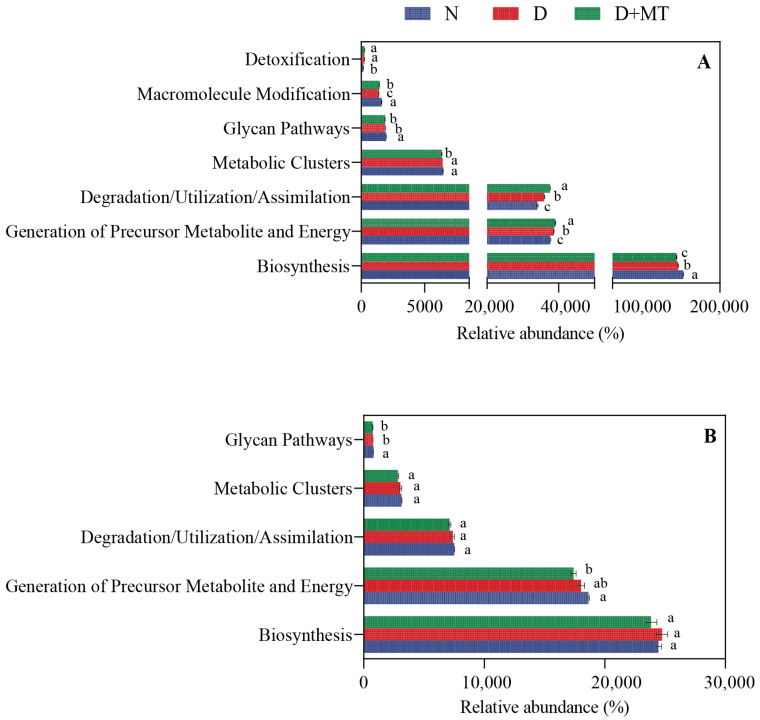
Functional abundance in bacterial and fungal communities at level 1 (**A**,**B**) and level 2 (**C**,**D**) under the combination of melatonin treatment and drought stress based on PICRUSt2. Data are reported as mean ± SE (*n* = 4). Different small letters indicate significant differences at the *p* < 0.05 level. N, normal control; D, drought stress; D + MT, melatonin + drought stress.

## Data Availability

The raw sequences have been deposited with NCBI under the BioProject accession number PRJNA873235.

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
