# Peer review of "Exogenous Melatonin Reprograms the Rhizosphere Microbial Community to Modulate the Responses of Barley to Drought Stress"

_ijms, 2022, doi:10.3390/ijms23179665_

Round 1
Reviewer 1 Report
Dear Authors,
I have a suggest to include few important markers related to drought (Leaf relative water content, Membrane stability index, H2O2, Proline, glycine-betaine and total chlorophyll). These are important markers/contents which helps to understand how soil application of melatonin has improved the above said parameters due to drought stress.
Regards,
K. N. Chandrashekara
Author Response
Overall comments: I have a suggest to include few important markers related to drought (Leaf relative water content, Membrane stability index, H2O2, Proline, glycine-betaine and total chlorophyll). These are important markers/contents which helps to understand how soil application of melatonin has improved the above said parameters due to drought stress.
Response : Thank you for the comments, we agree with the reviewer's suggestion that these drought-related markers are indeed important for monitoring the plant performance under drought stress. Therefore, the contents of H2O2, Proline, and Glycine-betaine in barley leaves were measured and the data were added. The related references were also cited in the text.
Reviewer 2 Report
The paper is interesting, it reports the exogenous application of melatonin to mitigate the effect of drought in barley plants. The results are interesting, they explain the modification of nonstructural carbohydrates, antioxidant enzymes, ABA, and the microbial community in the soil (bacteria and fungi) due to drought and the interaction between drought and the exogenous application of melatonin. However, I do have some comments for the authors on the document and the supplementary material presented which are listed below:
1.- Correct “and” in the list of authors.
2.- Delete the following sentence at the bottom of figure 1: Data are reported as mean ± SE (n = 4). Different small letters indicate significant differences at the P < 0.05 level.
3.- At the bottom of figure 2 describe if the letters A, B and C correspond to leaves, roots, and soil.
4.- In the supplementary material, the statistical literals are in descending order and in figure 2 they are in ascending order, please standardize.
5.- Please cite figures 3A and 3B (Chao1 and Shannon) in the text corresponding to subsection 2.3
6.- In the reference section, please italicize the scientific names of citations 20, 24, and 28. In addition, the following reference is not listed: T.D.; Bengtsson-Palme, J.; Callaghan, T.M.; et al. Towards a unified paradigm for sequence-based identifica-tion of fungi. Mol. Ecol. 2013, 22, 5271-5277.
Author Response
Response to Reviewer 2 Comments
Overall comments: The paper is interesting, it reports the exogenous application of melatonin to mitigate the effect of drought in barley plants. The results are interesting, they explain the modification of nonstructural carbohydrates, antioxidant enzymes, ABA, and the microbial community in the soil (bacteria and fungi) due to drought and the interaction between drought and the exogenous application of melatonin. However, I do have some comments for the authors on the document and the supplementary material presented which are listed below:
Response: Thank you for the comments, it has been revised accordingly.
Point 1: Correct “and” in the list of authors.
Response 1: It has been corrected.
Point 2: Delete the following sentence at the bottom of figure 1: Data are reported as mean ± SE (n = 4). Different small letters indicate significant differences at the P < 0.05 level.
Response 2: It has been deleted.
Point 3: At the bottom of figure 2 describe if the letters A, B and C correspond to leaves, roots, and soil.
Response 3: The information has been added.
Point 4: In the supplementary material, the statistical literals are in descending order and in figure 2 they are in ascending order, please standardize.
Response 4: They have been standardized.
Point 5: Please cite figures 3A and 3B (Chao1 and Shannon) in the text corresponding to subsection 2.3.
Response 5: It has been cited.
Point 6: In the reference section, please italicize the scientific names of citations 20, 24, and 28. In addition, the following reference is not listed: T.D.; Bengtsson-Palme, J.; Callaghan, T.M.; et al. Towards a unified paradigm for sequence-based identification of fungi. Mol. Ecol. 2013, 22, 5271-5277.
Response 6: The scientific names have been changed and the reference has been listed.
Round 2
Reviewer 1 Report
Dear Authors,
The revised version is fine.
Regards,
K. N. Chandrashekara